# Improved Real-Time Quaking Induced Conversion for Early Diagnostics of Creutzfeldt–Jakob Disease in Denmark

**DOI:** 10.3390/ijms24076098

**Published:** 2023-03-23

**Authors:** Remarh Bsoul, Eva Løbner Lund, Kimberley Burns, Mary Andrews, Neil McKenzie, Alison Green, Aušrinė Areškevičiūtė

**Affiliations:** 1Danish Reference Center for Prion Diseases, Department of Pathology, Copenhagen University Hospital, 2100 Copenhagen, Denmark; remarh.bsoul@regionh.dk (R.B.); eva.loebner.lund@regionh.dk (E.L.L.); 2National CJD Research and Surveillance Unit, Centre for Clinical Brain Science, University of Edinburgh, Edinburgh EH4 2XU, UK; kimberley.burns@nhs.scot (K.B.); mary.andrews3@nhs.scot (M.A.); n.mckenzie@ed.ac.uk (N.M.); alison.green39@nhs.scot (A.G.)

**Keywords:** Creutzfeldt–Jakob disease—CJD, improved real-time quaking-induced conversion—IQ, RT-QuIC, cerebrospinal fluid—CSF, prions, prion disease, neurodegenerative protein misfolding diseases, early diagnostics, recombinant PrP production and storage

## Abstract

Cerebrospinal fluid-based real-time quaking-induced conversion (CSF RT-QuIC) is currently the most prominent method for early detection of sporadic Creutzfeldt–Jakob disease (sCJD), the most common prion disease. CSF RT-QuIC delivers high sensitivity (>90%) and specificity (100%), which has been demonstrated by large ring-trial studies testing probable and definitive sCJD cohorts. Following the inclusion of CSF RT-QuIC in the revised European CJD Surveillance Network diagnostic criteria for sCJD, it has become a standard diagnostic procedure in many prion disease reference or surveillance centers around the world. In this study, we present the implementation of the second-generation CSF RT-QuIC (commonly known as Improved QuIC or IQ) at the Danish Reference Center for Prion Diseases (DRCPD). The method’s sensitivity and specificity were evaluated and validated by analyzing 63 CSF samples. These 63 samples were also analyzed at the National CJD Research and Surveillance Unit (NCJDRSU), based at the University of Edinburgh, UK; analysis was carried out using the first generation or previous CSF RT-QuIC method (PQ). The sensitivity and specificity of PQ during tests at the NCJDRSU were 92% and 100%, respectively. Using these 63 CSF samples, the agreement between the two RT-QuIC generations at DRCPD and NCJDRSU prion laboratories was 100%.

## 1. Introduction

Prion diseases are a family of invariably fatal neurodegenerative protein misfolding diseases caused by misfolding of cellular prion proteins (PrP^c^) into disease-associated isoforms, known as prions (PrP^sc^). Real-time quaking-induced conversion (RT-QuIC) is a method used to detect minute amounts (<10^−9^ g) of PrP^sc^ in cerebrospinal fluid (CSF) samples, using recombinant Syrian hamster full length PrP (rHaFLPrP, 23-231 aa) as a substrate. In the presence of pathological PrP, the recombinant PrP substrate undergoes a conformational change, which is detectable by increasing fluorescence emission [1]. 

This CSF RT-QuIC setup is referred to as first-generation RT-QuIC or Previous-QuIC (PQ), in the light of recent development of the second-generation RT-QuIC or Improved-QuIC (IQ).

Highly sensitive (>90%) and specific (100%) PQ has been increasingly applied for early diagnostics of sporadic Creutzfeldt–Jakob Disease (sCJD), the most common prion disease [2]; thus, in 2017, it was included as a criterion for sCJD diagnosis by the European CJD Surveillance Network [3]. Many more laboratories have adopted PQ for sCJD detection using CSF, and the reliability of the method has been proven by ring-trial studies [4]. The continuous effort to improve the RT-QuIC led to a second-generation RT-QuIC—the IQ. The IQ substrate is based on recombinant N-terminally truncated wild-type Syrian hamster PrP (rHaTrPrP, 90–231 aa), which has significantly improved the method’s sensitivity [5] and duration. Optimized IQ technical parameters, including enhanced kinetics, decreased assay time from ~90 to ~48 h [6]. 

Recent ring-trial studies of IQ performance for sCJD diagnostics have demonstrated the method’s high reproducibility and robustness [7]. Furthermore, the largest to-date retrospective population study of definitive sCJD cases demonstrated that IQ detected 80% of cases not detectable by PQ [8]. The study also evaluated IQ sensitivity for different sCJD subtypes and found it to be more sensitive for detection of rare disease subtypes, including VV1 (Parchi’s classification [9]), which have otherwise been reported as difficult to detect [10]. 

In recent years, RT-QuIC has also been tested for its applicability using different biospecimens (mainly olfactory mucosa [7,11] and skin biopsies [12]). Although they offer promising results, these studies require further validation. 

At the DRCPD, we have implemented and validated IQ for sCJD diagnostics. Our IQ validation process included *in-house*-produced protein stability experiments, and an evaluation of concordance between IQ results obtained in our laboratory and PQ results obtained at the National CJD Research and Surveillance Unit (NCJDRSU), based at the University of Edinburgh, UK. Currently, CSF PQ at NCJDRSU performs with 92% sensitivity and 100% specificity [13]. The same 63 CSF samples (probable sCJD: *n* = 18, negative sCJD: *n* = 45) were analyzed in both centers and the concordance of the results was 100%. In this paper, we describe our workflow and provide detailed information on the establishment of this highly sensitive and specific method; that is now used for early sCJD diagnostics also in Denmark. 

## 2. Results

### 2.1. rHaTrPrP Quality Assurance

To standardize our diagnostic workflow and assure the highest quality of each batch of *in-house*-produced rHaTrPrP, we established criteria for batch purity, stability, specificity, and sensitivity. rHaTrPrP purity is estimated by Coomassie SDS-page analysis, which must indicate a single band at 16.2 kDa and no clearly visible contaminants (Appendix A). 

rHaTrPrP stability, specificity and sensitivity are estimated by multiple unseeded and seeded reactions as described in Section 4.3. At DRCPD, produced rHaTrPrP must pass a >95% stability and specificity threshold to be used for diagnostic purposes. The batch of rHaTrPrP used for this study met all the set criteria.

We observed that different protein freezing strategies had an influence on protein stability. The best results were obtained when a batch was flash frozen with liquid nitrogen to avoid high salting and pH fluctuations, and stored at −80 °C. The results of this study were obtained using the same batch within 5 months of its initial quality evaluation. During this rHaTrPrP storage period, we observed a tendency in Relative Fluorescence Unit (RFU) decrease in sCJD brain homogenate (BH) controls. We tested if the RFU decrease over time was influenced by BH freshness; this was not the case, as even with a freshly prepared control, the RFU was lower with the older rHaTrPrP. 

### 2.2. Threshold for a Positive IQ

To determine the positivity of each well (replicate), a threshold for a cut-off value was established using 23 out of 45 negative (non-CJD) CSF samples. All 23 samples were analyzed in quadruplicates with both 15 µL and 30 µL CSF. Thus, 184 replicates were used in total to determine the mean RFU value. RFU was measured every 45 min for 50 h (65 data points), and the determined mean was 203 RFU with a standard deviation of 65.35. Therefore, a cut-off value for a positive replicate was set at 900 RFU, which approximately equals the mean RFU + 10 standard deviations.

### 2.3. Mean of Positive IQ Peaks

To define the mean RFU for a CJD-positive IQ, we used RFU values from the two best-performing (maximum RFU) replicates from each of the 18 CSF samples (30 µL) from patients with probable sCJD. A distribution pattern of the mean RFU is depicted in Figure 1a. All samples were significantly above the cut-off value. The distribution of all 18 CSF samples is depicted individually (Appendix A).

### 2.4. CSF Volume-Dependent IQ Sensitivity

We observed that IQ sensitivity depends on added CSF volume and calculated that our IQ sensitivity for 15 µL and 30 µL samples is 77% and 100%, respectively. Furthermore, the mean of RFU peaks is ~40% higher for 30 µL in comparison to 15 µL, giving a higher diagnostic certainty (Figure 1b). The volume-dependent sensitivity is likely related to the total amount of PrP^Sc^ in the sample, but it cannot be ruled out that other factors, such as relative salt concentration in CSF compared to that of the IQ mastermix composition, play a role. However, when considering all 72 replicates of the 18 samples with probable CJD, the mean curve for both 15 µL and 30 µL CSF volume still surpasses the set RFU cut-off value (Figure 1b). 

### 2.5. 100% Concordance between DRCPD and NCJDRSU

Our IQ results are based on a negative or positive outcome of a sample quadruplicate for any given CSF volume. We interpret CSF samples as described in Section 4.4. These criteria are identical to the PQ at NCJDRSU [4,14].

A total of 45 CJD CSF’s were concluded as negative at both DRCPD and NCJDRSU. At DRCPD, these samples were analyzed in quadruplicates with 15 µL and 30 µL of CSF, amounting to a total of 272 replicates. Only one of the 272 replicates gave a positive result. The IQ was repeated with the sample in question and yielded negative results, indicating that the formerly positive replicate was in fact false. Furthermore, 18 CSF PQ RT-QuIC positive samples identified at the NCJDRSU were also concluded to be positive at DRCPD using CSF IQ. Thus, the concordance between CSF RT-QuIC performance in both laboratories based on a total of 63 CSF samples was 100%. Detailed PQ and IQ results are summarized in Table 1.

## 3. Discussion

### 3.1. rHaTrPrP Validation, Preservation, and Stability over Time

The specific batch of rHaTrPrP used for this study was purified approximately 5 months before the completion of the validation. To our knowledge, there are no published reports describing rHaTrPrP stability over prolonged time of storage. However, to ensure the diagnostic quality, it is important to understand *in-house*-produced protein performance. We have observed a tendency in RFU decrease over prolonged storage time at −80 °C. This tendency was not affected by the freshness of the control sample. Therefore, the information about rHaTrPrP batch storage conditions and duration should be taken into consideration and clearly indicated, to establish a more standardized assay. Our current threshold for rHaTrPrP storage has been set to six months based on our validation period, though this shelf life could possibly be extended. The performance of different rHaTrPrP batches needs to be tested over a prolonged storage time to confirm our observations.

Lyophilization, which can often keep proteins stable for years, has been reported as a robust preservation technique for bank vole wild type PrP, eliminating the need for a refolding protocol [15]; it could also be relevant for storing other PrP types. Lyophilization is also reported successfully for RT-QuIC with various *alpha*-synuclein proteins [16,17].

### 3.2. IQ at DRCPD

Our findings confirm and further supplement the data published by other laboratories regarding the reliability and robustness of IQ as a method for the diagnostics of CJD. IQ differs from PQ substantially in the lag phase by showing a steep slope in RFU increase within the first 10–15 h of the assay time [8]. Due to this kinetic character, the assay time of IQ can be halved from ~90 h to ~48 h. In 14 out of 18 samples with probable CJD, we observed a general peak of RFU increase to 3000–6000 within 10–15 h. Only 4 out of 18 samples with probable CJD had both RFU peak at 15–20 h and relatively lower RFU at 2000–3000. This trend has been associated with certain CJD subtypes [8], but it is not possible to draw further conclusions regarding subtypes in this material due to lack of *post mortem* diagnosis confirmation. 

We also observed that IQ performed better with 30 µL CSF than with 15 µL CSF (Figure 1b). However, 4 out of 18 cases had higher RFU peak with 15 µL, which suggests that, in some cases, 30 μL CSF has a signal-suppressing effect due to overloading. The limited access to clinical data in the routine diagnostics means that we also cannot rule out that these 4 samples showing unusual tendencies have other suppressing factors, such as products of hemolysis. Our results using 15 μL and 30 μL CSF volume indicated that the analysis of both volumes simultaneously yielded a very reliable result, eliminating the need for extensive clinical data or additional CSF analyses. Therefore, we will be using both volumes in future diagnostics. Interestingly, in a previously published concordance study, a single laboratory performed IQ with 20 µL CSF and reported a detection of all positive samples included in the study [4]; this indicates that one volume may also be sufficient for reliable diagnostics. It would be interesting to test whether we could achieve the same sensitivity with our IQ and the current cohort with a 20 uL CSF set-up, which would optimize the use of CSF samples from 180 µL to 80 µL. 

We aimed to achieve >95% concordance between CSF IQ RT-QuIC performed at DRCPD and PQ RT-QuIC performed at NCJDRSU, which has reported PQ sensitivity and specificity of 92% and 100%, respectively. These parameters were established based on analyses of >1600 CSF samples, using current criteria for the interpretation of results [13]. The results of this study of 63 CSF samples show complete agreement between these two RT-QuIC methods regarding the identification of probable sCJD and non-CJD CSF samples. 

### 3.3. Uniform IQ Controls

A challenging aspect of prion disease research is the scarcity of required CJD biospecimens. BH from CJD patients or pre-fibrillated recombinant PrP, are currently standardized positive controls for CSF RT-QuIC [14]. However, controls and samples being prepared differently introduce certain limitations. Examples of such limitations include differences in mastermix composition, and volumes used for CSF sample and BH control preparation. Alternatively, prions from BH or pre-fibrillated recombinant PrP could be spiked in a non-CJD CSF sample, to obtain a positive control that resembles a patient sample and makes sample and control mastermix preparations more uniform. This approach, however, requires continuous access to CSF samples from healthy donors and availability of relevant clinical information. Thus, the use of a synthetic universal medium would potentially be of great importance for uniformity of RT-QuIC set-up, locally and internationally.

### 3.4. Concluding Remarks

RT-QuIC has proven to be a unique tool for early CJD detection. IQ is so far the most advanced RT-QuIC assay, given the overall higher sensitivity and shorter assay time. In this study, we presented the implementation and validation of IQ at the Danish Reference Center for Prion Diseases, and report a 100% performance concordance with PQ at the National CJD Research and Surveillance Unit, based at the University of Edinburgh, UK. 

## 4. Materials and Methods

### 4.1. Production of Syrian Hamster Truncated Recombinant Prion Protein (rHaTrPrP) 90-231 aa

rHaTrPrP was produced *in-house* using published protocols as guidelines [1]. More specifically, high-copy number plasmid PRSET (Invitrogen^®^, Dreieich, Germany) was commercially ligated (Azenta^®^, Leipzig, Germany), with DNA sequences corresponding to the prion protein gene (*PRNP*) sequence encoding amino acids from 90 to 231 in the Golden Syrian hamster (PDB DOI: https://doi.org/10.2210/pdb1B10/pdb, accessed on 20 March 2023). Ligation was performed to strategically bypass the N-terminal *His*-tag of the PRSET plasmid. BL(21)E3 chemically competent cells were transformed, and appropriate colonies were selected for glycerol stocks. A tip of glycerol stock was used to inoculate 5 mL LB media at 37 °C, 220 rpm, 5–7 h. The LB media-containing cultures were diluted 1:500 in 4 × 500 mL auto-induction TB media, and incubated at 37 °C, 200 rpm, overnight, in non-baffled flasks. Cells were pelleted at 3200× *g*, 10 min, RT, and divided into six 50 mL tubes. Cell pellets were washed with 2× BugBuster Mastermix (15 mL/10 g wet cells) using mild homogenization and allowed to gently mix by rotation for 10–30 min, RT. Cells were then pelleted at 13,000× *g,* and the above-described extraction was repeated 2 times with 10% BugBuster Mastermix. The resulting white pellets were totally solubilized in denaturing buffer (8 M Gnd-HCl, pH 8) (30 mL/cell pellet) by rotation for 24–72 h, 10 rpm, 4 °C. Ninety mL (total 180 mL in 50% ethanol suspension) of Ni-NTA Superflow (Qiagen®, Kista, Sweden) was washed with a denaturing phosphate buffer (6 M Gnd-HCl, 100 mM sodium phosphate, 10 mM Tris, pH 7.8) by rotation at 10 rpm, 3 × 10 min, followed by centrifugation at 1500× *g* and discarding of the supernatant. The resulting slurry was equilibrated with the rHaTrPrP solution and allowed to bind by rotation at 10 rpm, 1 h, RT. The slurry was then pelleted and packed in a suitable column onto an ÄKTA FPLC, at 10 mL/min, 10 min. rHaTrPrP was folded with a linear gradient (100 mM sodium phosphate, 10 mM Tris, pH 7.8) over 200 min at 5 mL/min flow. The product was eluted with a linear imidazole gradient (100 mM sodium phosphate, 10 mM Tris, 500 mM imidazole, pH 6) over 45 min at 6 mL/min and collected as a broad peak between 250–350 mM imidazole. The elute was sterile filtered (0.2 µm), while the concentration was estimated with spectrophotometer at Abs280 (250 mM imidazole as base line) and subsequently diluted to 0.8 mg/mL with a 10 mM phosphate buffer, pH 6.5. The elute was dialyzed extensively against a 10 mM phosphate buffer at pH 6.5 (>20 times the elute volume), and the final product concentration was estimated by Qubit^TM^ 4 fluorometer using protein assay. The batch is then properly aliquoted on ice, flash frozen with liquid nitrogen and stored at −80 °C. The yield of the batch used for this study was ~70 mg. The typical range over four batches was 40–70 mg. 

### 4.2. Coomassie SDS-Page Analysis

To estimate the purity of produced rHaTrPrP, samples with a total volume of 20 µL (5 µL rHaTrPrP elute, 5 µL 4× Loading Buffer, 2 µL 10× Sample Reducing Agent, 8 µL sterile water) were heated at 85 °C for 5 min and loaded on a 10-well 10%, Bis-Tris, 1.0 mm gel. The gel was then stained with 200 mL Coomassie stain (2.4 g/L brilliant blue, 60% methanol, 12% glacial acetic acid) for 15 min, 300 rpm, in microwaved (1 min) stain solution. De-staining (10% methanol, 10% glacial acetic acid) was repeated twice for 15 min, 300 rpm, in microwaved (1 min) 200 mL de-stain solution. Stained gel was visualized on ChemiDoc^TM^ imaging platform with stain-free settings. 

### 4.3. HaTrPrP Stability and Sensitivity Evaluation

The stability of newly produced rHaTrPrP was evaluated by plate setup consisting of 32 wells of unseeded mastermix, Alzheimer’s Disease (AD) BH and sCJD BH at a dilution of 10^−7^. The threshold for deviation from expected results was set at <5%, i.e., 1/32 reactions could yield a false positive or negative result. The sensitivity of the rHaTrPrP produced following the protocol described in Section 4.1 was evaluated by an endpoint dilution series of a 10% *w*/*v* sCJD BH ranging from 10^−5^–10^−15^. This was only performed for the initial batch and is not included in our batch-to-batch evaluation.

### 4.4. IQ RT-QuIC Plate Outline, Instrument Parameters and Interpretation of Results

Each black, 96-well plate with a flat and transparent bottom, was loaded with unseeded mastermix, sCJD BH, AD BH, a non-CJD and probable CJD CSF samples (15 µL and 30 µL). All controls and samples were analyzed in quadruplicates. The plate was then sealed with optical adhesive film and incubated in an Omega FLUOstar^TM^ (BMG^®^ LABTECH) plate reader with continuous cycle of 60 s. shaking, at 700 rpm, and 60 s. resting, for total of ~48 h, at 55 °C. A fluorescence measurement, expressed in RFU at 450 ± 10 nm excitation and 480 ± 10 nm emission, was taken every 45 min. Gain was set to 1000. 

IQ results for a quadruplicate setup were interpreted as either negative (0/4), inconclusive (1/4) and repeated, or positive (2–4/4) [4]. 

### 4.5. Brain Samples and Brain Homogenate Control Preparation

At DRCPD, samples collected from certain brain areas, i.e., frontal cortex (FC) and cerebellum (CB), were immediately preserved by freezing them at −80 °C. The rest of the brain was formalin-fixed (10% neutral buffered) for a month, treated with formic acid for 1 h and paraffin-embedded. The definitive diagnosis was provided based on prion detection by immunoblotting, *PRNP* sequencing, tissue morphology and immuno-histochemistry. Frozen brain samples with a *post mortem*-confirmed diagnosis of sporadic CJD MM1 and AD were used for the preparation of a positive and negative BH control, respectively. AD cases were confirmed to be negative for prions by immuno-histochemistry. BH controls were prepared as a 10% *w*/*v* in TBS-based extraction buffer and homogenized using Dispomix^TM^ tissue homogenizer for 15 s, 4000 rpm. The homogenates were then aliquoted to 100 µL and stored at −20 °C. The freeze–thaw cycles of each control were recorded; these cycles did not occur more than five times. BH were serially diluted to 10^−15^ in sterile filtered (0.2 µm) 0.1% SDS in PBS, at pH 7.4, to determine the best performing concentration, which subsequently was used as a control for diagnostic IQ.

### 4.6. CSF Samples

In total, 63 CSF samples (*ante mortem*) were included in this IQ validation. Overall, 48 of these samples were delivered to DRCPD for diagnostic purposes from several neurological departments in Denmark. Samples were shipped on dry ice and, upon arrival, stored at −80 °C. The remaining 15 samples were from patients with probable sCJD and provided by NCJDRSU [3]. Additional information regarding RT-QuIC generation (PQ), CSF volume (*n* = 14 at 30 µL, *n* = 1 at 15 µL) and RFU for the two best-performing replicates for each sample was provided.

### 4.7. IQ Mastermix Preparation

All Mastermix buffers and solutions were freshly prepared on the day of each RT-QuIC assay, sterile filtered (0.2 µm) and kept cool until use. rHaTrPrP aliquots were thawed at RT and moved to 4 °C. The recombinant protein was spin filtered through a 100 kDa MWCO at 4000× *g*, 4 min, 4 °C, quantitated using Qubit^TM^ 4 fluorometer protein assay and handled without vortexing; it was mixed only by tube flipping. Each mastermix was prepared in a separate tube and kept at 4 °C until dispersed. BH controls (100 µL total volume/well) were composed of a 98 µL mastermix (10 mM PBS at pH 7.4, 300 mM NaCl, 1 mM EDTA at pH 8.0, 10 μM ThT, 0.1 µg/µL of rHaTrPrP) and 2 µL serially diluted pre-vortexed BH (10^−7^). For unseeded controls, 2 µL 0.1% SDS in PBS at pH 7.4 were added instead of BH. CSF samples (100 µL total volume/well) were composed of a 70 µL or 85 µL mastermix (10 mM PBS at pH 7.4, 300 mM NaCl, 1 mM EDTA at pH 8.0, 10 μM ThT, 0.002% SDS, 0.1 µg/µL of rHaTrPrP), and 15 µL or 30 µL neat pre-vortexed CSF, respectively. 

### 4.8. Software and Statistical Data Analysis

FLUOstar^TM^ Omega Plate Reader (BMG^®^ LABTECH) was connected to Omega Control and Omega MARS^TM^ software for instrument control and data analysis, respectively. Using Omega MARS^TM^, multiple plates were merged into one layout, and all data were exported as a table sheet of all cycles for statistical data analysis in GraphPad^®^ Prism^TM^ 9.

A cohort of non-CJD CSF samples was used to establish a RFU cut-off value for a positive sample. A mean (*µ*) of all replicates and standard deviation (*σ*) for all measured cycles, except 0 min, was calculated. The threshold for a positive CSF was set at the RFU corresponding to +10 standard deviations. Confidence Interval (95%) was based on the standard error of the mean of all replicates.

## Figures and Tables

**Figure 1 ijms-24-06098-f001:**
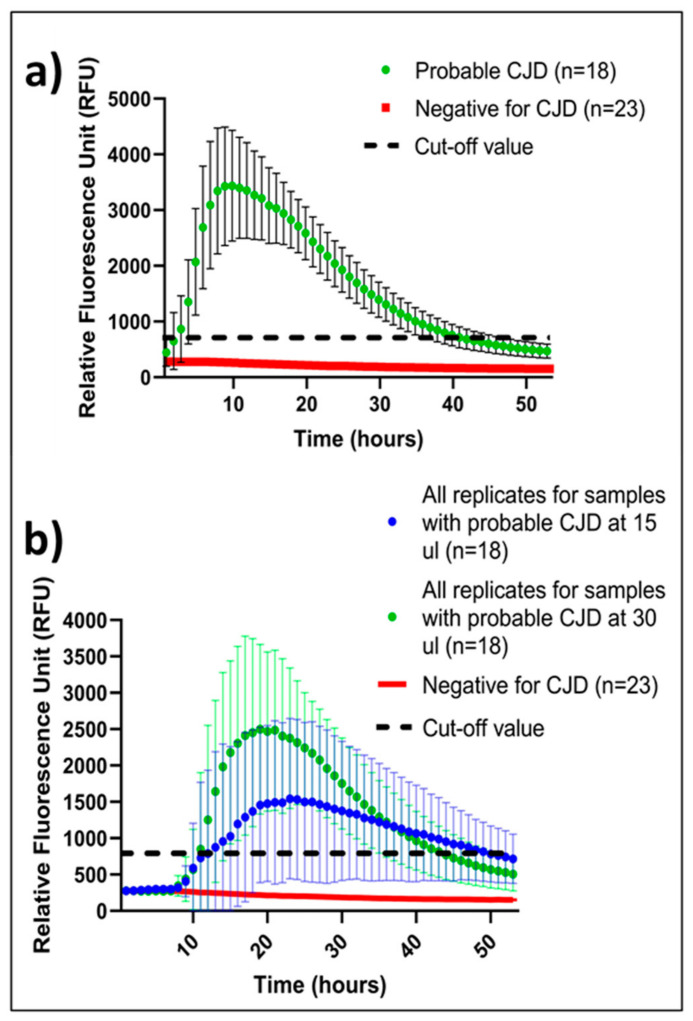
(**a**) IQ kinetics, time, and intensity with 30 µL CSF. Green is a mean of the two best replicates in a quadruple setup with error bars (95% confidence interval). Red is a mean of all negative replicates in a quadruple setup. Dotted line represents the RFU cut-off value for a positive replicate. (**b**) CSF volume-dependent IQ kinetics, time, and intensity. Blue and Green curves represent mean RFU of 15 μL and 30 μL replicates (*n* = 144) of CSF samples from probable sCJD patients. Error bars depicts 95% confidence interval. Red curve represents a mean of quadruplicates of 23 CSF samples from non-CJD patients tested at 15 µL and 30 µL. Dotted line represents the RFU cut-off value for a positive replicate.

**Table 1 ijms-24-06098-t001:** Overview of CSF IQ results at DRCPD and PQ results at NCJDRSU, where 18 probable CJD CSF samples were analyzed in a quadruplicate setup with 15 µL and 30 µL. Red: Negative, Yellow: Inconclusive, Green: Positive.

	DRCPD		NCJDRSU	
**Probable CJD (*n* = 18) Sample No.**	Positive Replicate/ Total No. of Replicates; Sample Volume 15 µL	Positive Replicate/ Total No. of Replicates; Sample Volume 30 µL	Positive Replicate/ Total No. of Replicates; Sample Volume 15 µL	Positive Replicate/ Total No. of Replicates; Sample Volume 30 µL
1	4/4	4/4	4/4	4/4
2	2/4	4/4	4/4	4/4
3	2/4	4/4	2/4	2/4
4	0/4	2/4	4/4	4/4
5	1/4	4/4	3/4	1/4
6	2/4	4/4	4/4	4/4
7	4/4	4/4	3/4	4/4
8	4/4	4/4	4/4	4/4
9	1/4	3/4	3/4	1/4
10	1/4	4/4	1/4	3/4
11	2/4	4/4	4/4	4/4
12	4/4	3/4	4/4	3/4
13	3/4	4/4	3/4	4/4
14	4/4	4/4	4/4	4/4
15	3/4	3/4	4/4	4/4
16	3/4	2/4	4/4	3/4
17	4/4	4/4	3/4	3/4
18	2/4	3/4	1/4	4/4

## Data Availability

Not applicable.

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
