# Peer review of "Improved Real-Time Quaking Induced Conversion for Early Diagnostics of Creutzfeldt–Jakob Disease in Denmark"

_ijms, 2023, doi:10.3390/ijms24076098_

Round 1

Reviewer 1 Report

The manuscript describes a direct comparison of the use of QuIC for detection of prions in CSF of a small number of CJD patients and negative controls using either full-length of truncated hamster substrate. 
The number of samples is very small to confirm efficacy although consensus between the two methods and an external lab is provided. The study is useful to other laboratories using similar methods to provide confidnece in the utility of use of the truncated substrate.

I have a couple of specific questions:

The authors present different RFU readings depending on the volume of CSF and comment on this ias a parameter of sensitivity. Could this factor be proportional to relative salt concentration in the sample due to addition of CSF rather than the mastermix?

On page 9. line 166 te authors suggest overloading in some cases where 15ul in 4 cases produced higher RFU peaks. Could this be a factor of some specific contaminant in these specific samples such as products of hemolysis, which are inhibitory? Are any measurements for hemolysis measured to ensure the equivalency of samples in this regard?

I wasn't sure of how the section of discussion on uniform IQ controls fit in - were any of these kind of controls presented in the data?

A number of grammatical errors were present throughout the manuscript. Careful editing is recommended.

Author Response

The authors present different RFU readings depending on the volume of CSF and comment on this ias a parameter of sensitivity. Could this factor be proportional to relative salt concentration in the sample due to addition of CSF rather than the mastermix?

- This cannot be ruled out.  I have added your observation at line 127.

On page 9. line 166 te authors suggest overloading in some cases where 15ul in 4 cases produced higher RFU peaks. Could this be a factor of some specific contaminant in these specific samples such as products of hemolysis, which are inhibitory? Are any measurements for hemolysis measured to ensure the equivalency of samples in this regard?

- Hemolysis could be a suppressing factor such as high blood cell count is known to be. Currently we ask for information on red blood cell count and total protein count only, but not for data on hemolysis. I have extended the discussion at line 170 with this in mind.

I wasn't sure of how the section of discussion on uniform IQ controls fit in - were any of these kind of controls presented in the data?

- We noticed when setting up the RT-QuIC in our laboratory, and when doing validation studies, that the current brain control system is problematic and limiting the RT-QuIC use to a biosafety level 3 laboratories. We think it is a relevant and important consideration to take into account when setting up RT-QuIC analysis and thus we want to share our insights with readers and make a point that currently applied controls could resemble the sample master mix more, limit the use of fresh brain samples, and streamline the RT-QuIC workflow even more.  We are currently exploring different solutions for such control set-up, but the data is not sufficient yet.

A number of grammatical errors were present throughout the manuscript. Careful editing is recommended.

- Thank you pointing that out. We apologize and will make sure to correct it

Reviewer 2 Report

In this article Bsoul et al describe different experiments for the implementation of RT-QuIC IQ on the Danish CJD surveillance program. To fulfill this objective, the authors performs a double test to contrast their results with the National CJD Research and Surveillance Unit (NCJDRSU) of United Kingdom. This unit have got an stablished and standardized method for CJD diagnosis using RT-QuIC IQ. The manuscript is very clear and straightforward, proving the suitability of this in vitro technique for the diagnosis of CJD using CSF of suspected cases. However, there is some minor issues that should be addressed by the authors:

1 - In figure S1 the authors show a result of a Coomassie SDS-page gel. The gel is very clear and shows the purity of the protein and the absence of other peptide contaminants. However, there is no indication of what has been loaded in each well, so it is not possible to fully understand the experiment. Maybe is a mistake related with the journal or anything, but this information should be included in the figure, even if it is supplementary material. The figure is exactly the same as the Original image of Blot, so maybe there were some mistake when uploading it or something. In any case, this issue must be fixed.

2 - In the line of the aforementioned problem with figure S1, there is a reference in the manuscript about the figure S2, but there is not figure S2 in the supplementary material. Again, it looks that there is some kind of mistake regarding these figures.

3 - In lines 100-101 the authors say: "The distribution of all 18 CSF samples is depicted individually in supplementary figure 1 (Fig. S2)". Again, it looks like there is some kind of error regarding supplementary material.

4 - In lines 77-79 authors say: "A criterion of >95% stability and specificity mus be met. The batch of rHaTrPrP used for this study met all set criteria." These criteria are explained in Material and methods and it is specified that Mastermix alone, ADBH and sCJD BH are used as control. However, there is no indication of the dilution of ADBH and sCJD BH used. This information could be included for more clarity.

5 - In HaTrPrP stability and sensitivity evaluation (Material and methods) the authors indicate a test using sCJD BH 10% diluting ranging 10-5 to 10-15 to test sensitivity. However, the result of the batch used in the manuscript is not shown. It would be interesting to add this information to the manuscript to know exactly the level of detection of the batch of protein used for the RT-QuIC IQ.

6 - CSF samples were antemortem or postmortem? This information is important and should be included in Material and methods. It is interesting to know in what stage of disease the CSF samples were taken.

Author Response

1 - In figure S1 the authors show a result of a Coomassie SDS-page gel. The gel is very clear and shows the purity of the protein and the absence of other peptide contaminants. However, there is no indication of what has been loaded in each well, so it is not possible to fully understand the experiment. Maybe is a mistake related with the journal or anything, but this information should be included in the figure, even if it is supplementary material. The figure is exactly the same as the original image of Blot, so maybe there were some mistake when uploading it or something. In any case, this issue must be fixed.

1) It seems that you have not received all uploaded figures. We do have figures with all annotations you are concerned about. I will inform the editorial office about this mistake.   If the supplementary information cannot be retrieved from the Zenodo link then I can upload it a zip folder elsewhere. I will contact the editorial office regarding missing suppl. files. We apologize.

2 - In the line of the aforementioned problem with figure S1, there is a reference in the manuscript about the figure S2, but there is not figure S2 in the supplementary material. Again, it looks that there is some kind of mistake regarding these figures

2) I had uploaded a zip folder through Zenodo (https://zenodo.org/deposit/7640195) which contain the data asked for. If the supplementary information cannot be retrieved from the Zenodo link then I can upload it a zip folder elsewhere.

3 - In lines 100-101 the authors say: "The distribution of all 18 CSF samples is depicted individually in supplementary figure 1 (Fig. S2)". Again, it looks like there is some kind of error regarding supplementary material.

3) Thank you for finding that mistake. It is corrected in the text lines 99.

4 - In lines 77-79 authors say: "A criterion of >95% stability and specificity mus be met. The batch of rHaTrPrP used for this study met all set criteria." These criteria are explained in Material and methods and it is specified that Mastermix alone, ADBH and sCJD BH are used as control. However, there is no indication of the dilution of ADBH and sCJD BH used. This information could be included for more clarity.

4) I have added this information in line 253.

5 - In HaTrPrP stability and sensitivity evaluation (Material and methods) the authors indicate a test using sCJD BH 10% diluting ranging 10-5 to 10-15 to test sensitivity. However, the result of the batch used in the manuscript is not shown. It would be interesting to add this information to the manuscript to know exactly the level of detection of the batch of protein used for the RT-QuIC IQ.

5) The sensitivity end point dilution test was done initially to evaluate the quality of our in-house production, but we do not do this batch-to-batch evaluation. I have re-edited at line 252-255 to make that obvious. The batch used for the validation period was the 2nd batch in the production line.

6 - CSF samples were antemortem or postmortem? This information is important and should be included in Material and methods. It is interesting to know in what stage of disease the CSF samples were taken.

6) information regarding the ante-mortem sample collection has been added to line 284. I agree that disease onset and more clinical data would be interesting to include, however,  that would require access to patients’ clinical information from different neurology departments in Denmark and involvement of many clinicians, which is difficult to do considering retrospectively collected samples. We hope to be able to collect more clinical information on the samples delivered now and in the future.